# Can big data increase our knowledge of local rental markets? A dataset on the rental sector in France

**Guillaume Chapelle**[1,2]*, **Jean Benoît Eyméoud**[3,2]

**1** CY Cergy Paris Université, THEMA, CNRS, Cergy, France, **2** LIEPP, Sciences Po, Paris, France, **3** Banque de France, Paris, France

☉ These authors contributed equally to this work.

* guillaume.chapelle@cyu.fr

**Data Availability Statement:** ALL the reproduction programs are available in the Open Science Framework Repository (DOI: 10.17605/OSF.IO/CW37U) The repository also contains aggregate data on local rental market necessary to reproduce

## Abstract

Social Scientists and policy makers need precise data on market rents. Yet, while housing prices are systematically recorded, few accurate data sets on rents are available. In this paper, we present a new data set describing local rental markets in France based on online ads collected through to webscraping. Comparison with alternate sources reveals that online ads provide a non biased picture of rental markets and allow coverage of the whole territory. We then estimate hedonic models for prices and rents and document the spatial variations in rent-price ratios. We show that rents do not increase as much as prices in the tightest housing markets. We use our dataset to estimate the market rent of each transaction and of social dwellings. In the latter case, this allows us to estimate the in-kind benefit received by social tenants which is mainly driven by the level of private rent in their municipality.

## Introduction

Having precise knowledge of local rental markets has been a growing interest for policy makers and researchers. For example, while housing bubbles are a recurrent concern of macro-prudential authorities [1], these phenomena can appear locally as in Paris in 1991 and their identification requires data on both local housing prices (selling price) and returns (rental price). Moreover, having precise knowledge of local rents might be important for taxation purpose. In France, local taxes such as housing or property taxes should be based on the rental value of dwellings. However, the current tax base was estimated in the 1970s and revisions implemented ever since have been homogeneous. As the divergence between local markets or even neighborhoods has not been accounted for, this has led to a strong discrepancy between the tax base and the true market value with significant redistributive consequences [2, 3]. The 2020 Finance Law provides that rental values have to be reviewed before 2026, requiring the collection of additional data for this purpose. More broadly, rental data might also be needed to assess and compare the cost of living in different cities or neighborhoods [4].

the tests for the bias (Table 5). Underlying micro data can be accessed by submitting a request to the LIEPP, Sciences Po (liepp@sciencespo.fr). Enquête Logement data underlying Table 1 can be accessed submitting a request on http://quetelet. progedo.fr/. DVF+ files are available from the OSF repository and can also be accessed from https:// datafoncier.cerema.fr/donnees/autres-donnees-foncieres/dvfplus-open-data. Access to RPLS data including information on rents is restricted. Researchers from public institutions can access the version used submitting a request to rpls. cgdd@developpement-durable.gouv.fr.

**Funding:** The authors acknowledge the support from ANR-11-LABX-0091 (LIEPP) and ANR-11-IDEX-0005-02 awarded to GC and JBE. The funders had no role in study design, data collection and analysis, decision to publish, or preparation of the manuscript.

**Competing interests:** Banque de France is not a commercial company but an independent public institution that has been a member of the Eurosystem of central banks since 1999. It had no role in study design, data collection and analysis, decision to publish, or preparation of the manuscript.

Between December 2015 and January 2018, we periodically collected, cleaned and analyzed housing rental ads from the two largest French real estate websites. These two websites were the leaders in the market with a monthly stock of ads on the rental market oscillating between 500,000 and 750,000 [5–7]. Each ad provides the location of the housing good as well as its hedonic characteristics, offering the possibility to describe local housing markets and to estimate local rent indices. In this paper we describe the method used to collect these online data and describe the data-set. We then confront these data with a more conventional data collection method. We do not find any significant differences between the rent measured from online ads and the average rent measured with surveys. We estimate hedonic indices for prices and rents and systematically document the spatial variation in the rent-price ratio. We show that rents does not increase as much as prices in the tightest housing markets. Lastly, we use our dataset to estimate the market rent of each transaction and social dwellings. In the latter case, market rent can be used to measure the in-kind benefit received by social tenants which is mainly driven by the level of private rent in the municipality.

## Background

### Lack of data on the French rental market

In France, housing prices are recorded by the fiscal administration as the transaction is taxed. However, the rental market is currently mainly studied by the National Statistical Institute which produces the French Housing Survey [8] and the Survey on Rents and Housing Expenditures [9]. Both provide good quality data on the rental sector but they have two drawbacks. First, they are only representative at the national level as they have a limited number of observations. The French housing survey has 36 000 households but only 2 947 tenants in the private sector, the Survey on Rents and Housing Expenditures 4300 households. Thus, they cannot be used to monitor the rental dynamics of a city or an urban area. Second, they do not allow monitoring of the market of new leases, but are representative of the whole rental sector where rent revision is regulated. Another dataset exists at the family branch of Social Security, which is collecting information provided by recipients of housing allowances. However, this dataset has limited use, as it does not provide characteristics of the dwelling besides the municipality and the rent.

This lack of information at the local level led to three initiatives: the Observatoire des Loyers de l'Agglomération Parisienne (OLAP) [10], followed by local observatories, Observatoires Locaux des Loyers (OLL) [11] supported by the French Ministry of Housing. In parallel, Connaître Les Loyers et Analyser les Marchés sur les Espaces Urbains et Ruraux (CLAMEUR) [12] was also created. The OLAP is publicly supported and was first in charge of observing rents in the urban area of Paris while progressively extending its survey to the main French urban areas. It produces two micro level data sets: a panel data set and a time series of yearly cross sectional observations from 1990 until nowadays. Even though these data sets are of good quality, they also present two main limits: they only cover a limited share of the French territory and their access to researchers appears relatively difficult. To our knowledge there exists only a single published study based on this data set [13]. More recently, new local rent observatories have been developed in several urban areas. However, the number of urban areas covered remains limited as there are only 35 local observatories, including the OLAP in Paris. The third initiative, CLAMEUR, collects rental data from real estate agents and insurance companies. It provides a yearly average of the rent per square meter for about 887 French municipalities and groups of municipalities (Etablissements Publics de Coopération Intercommunale). If such a source provides useful information on local markets condition and their dynamics, few details concerning the variables available are provided in their databases. To our knowledge no

academic paper has ever used their micro-level data. Moreover, their data have limited geographical coverage (887 out of 36 000 municipalities). Lastly, many websites provide estimates of local rents and prices but their estimates are heterogeneous and suffer from methodological opacity [14].

These aforementioned sources might present some leads to dealing with the limited knowledge of local rental markets. However, these surveys or administrative database require an important and potentially costly treatment to increase the number of observations (for the survey-based data) or the number of variables (for the administrative data) and might not be available to researchers. On the other hand, online ads can present a promising way to document the variations in local rents as illustrated in the US with Craigslist [15].

## The growing coverage of real estate websites on the housing market

Housing surveys show that an increasing share of private tenants find their accommodations on real estate websites. Nowadays a vast majority of private landlords or real estate agencies use the internet to find tenants as illustrated in Table 1. Even if these channels do not constitute the whole market, as 22% of the tenants found their flat by alternate channels. Namely 19% by word of mouth, 1% from the employer and 2% from social services. We may be able to observe the vast majority of the market. From our perspective, exploiting housing advertisements posted online can provide an interesting and complementary way to survey local housing markets at a moderate cost.

## The growing role of user-generated content in research

Online data and ads have been increasingly used in research, particularly to study the real estate market. For example, a series of papers have exploited online ads to investigate the impact of energy efficiency labels on housing prices [16, 17]. Other papers used these data to study the impact of rent control [18], describe the impact of Airbnb [19–21] or evaluate the incidence of taxation [22]. Ads can also be used to follow real estate dynamics [23] or to compare housing prices across countries [24]. However, while online data are increasingly popular for studying real estate market dynamics, extremely few papers have studied their reliability [14, 15].

One key advantage of online data is their reduced cost, the high number of observations available and their high coverage and granularity when compared with standard survey data. In 2012, the total budget dedicated to the French Local Rent Observatories to follow 19 urban areas was 2.5 Million euros [25, 26]. Although webscraping can allow the gathering of a large number of observations with a homogeneous method and for a limited cost, it is not always

**Table 1. Method used to find a flat in the rental sector (%).**

|  | Not Furnished | Furnished | Total |
|---|---|---|---|
| Privately (ads on internet or Newspapers) | 37 | 42 | 37 |
| Real Estate Agency | 41 | 22 | 39 |
| by word of mouth | 19 | 20 | 19 |
| From the employer | 1 | 3 | 2 |
| Social Services | 2 | 10 | 3 |
| Others | 0 | 3 | 1 |
| Total | 100 | 100 | 100 |

*Source:* Author's computation from the French Housing Survey 2013 (INSEE). Households in the private rental sector installed for less than 4 years.

possible to access these datasets easily as public APIs are not always available and data might also be protected by copyrights. Nevertheless, this paper documents the reliability of online ads that can be accessed through specific agreements with online platforms.

## Materials and methods

### Main dataset

**Scraping process.** There are several real estate websites in France. To gain access to the biggest source of data we decided to focus on the two largest. We used their public API which displayed no restriction of use. The first had about half of its posts from landlords and half from real estate agents. The second had mostly ads from real estate agents. The information we wanted to extract consisted of a set of posts available on the rental websites. Each ad has a unique identifier, pictures, a short text describing the offer and a standardized table presenting the most important characteristics, such as the surface, the number of rooms, the monthly rent, or the type of contract (furnished or not). It is also localized thanks to the name of the municipality, a zipcode and a map indicating the geographic coordinates which can be more or less precise (city level, neighborhood or address). The non-structured part of the ad (description) allows us to identify key words in order to find additional information, such as the presence of an elevator, the storey, and the amount of extra expenditures.

To obtain the data from the two websites, we created programs that sent requests for every municipality in France to their public API and stored the information sent in a dataset. We then cleaned the dataset and structured it to yield a structured format for each post. Lastly, we saved the database in comma-separated values format or SQL database. Overall, the operation took between 10 hours and 2 days, depending on the website and the period of time. We repeated the process of scraping every month for each website from December 2015 until January 2018 and ended up with a database of 4.3 million posts from the rental sector.

**Cleaning the data.** The cleaning procedure starts by identifying repeated posts that had the same identifier between each wave. We also identify similar posts between both sites using the post's description. We keep only one observation per ad and keep the number of occurrences of the post. Other papers [23] use a machine learning algorithm to identify similar ads with different identifier. In our approach, we generate a full set of variables describing each housing unit and consider that a unit with the same price and the same characteristics (surface, number of rooms, amenities and geocoding) posted in the same month are duplicates. Once the duplicates dropped, for the sake of comparability with the local observatories data, we follow the recommendation of the National Rent Observatory and keep observations with the rent between 80 and 15 000 euros and the surface between 5 and 500 square meter. As the main point of our study is to have geolocalized data, we keep only observations that provide a city name or a precise geographic location. This procedure creates the final database, which we describe in the next section. In second part of the paper, hedonic models are then estimated removing also the outliers based on the rent per square meters (see S7 Table in S1 Appendix).

Overall, our cleaning procedure decreases the number of observations by 12.87%. The largest part of this decrease is explained by observations that do not report a surface. We believe that the price per square meter is the relevant parameter to characterize the housing market, as it provides a rental value used in other countries and is easily comparable.

**Creation of the variables.** Overall our cleaned database has about 4.3 millions observations collected between December 2015 and January 2018. The following section describes the different variables and their origins.

*The average rent per square meter.* Online ads directly provide the location of the dwelling. A total of 60% of the ads are located at the broadest level (French municipalities) while 40%

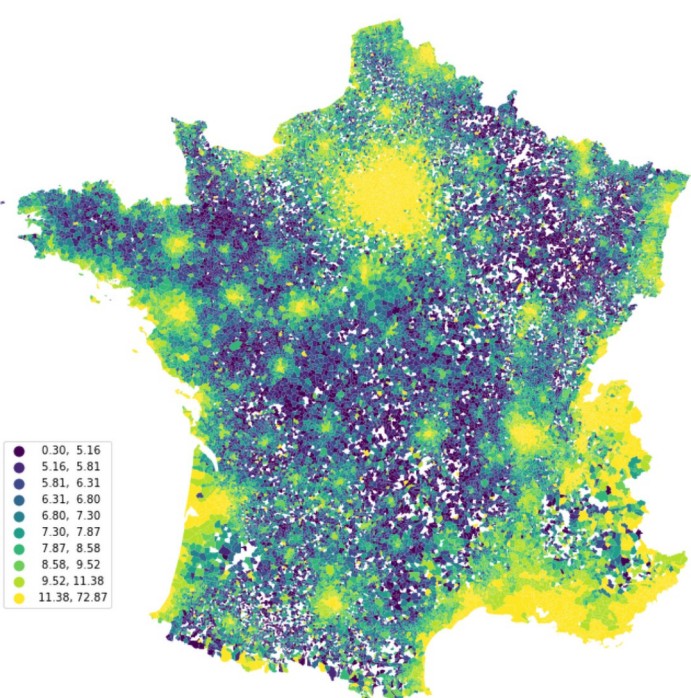

**Fig 1. Average gross rent in French municipalities.** *Note:* Author's computations and ADMIN EXPRESS COMMUNE which is under an Open Licence Etalab https://www.data.gouv.fr/fr/datasets/admin-express/.

remaining are precisely geocoded at the address or neighborhood level using the information provided by the user or the location of the device used by when creating the ad. This database thus provides fine grain data, as even municipalities remain quite small [4]. This allows us to compute the average gross rent per square meter for the majority of the municipalities in France as illustrated in Fig 1 and presents easy identification of the main urban areas and the places close to the frontiers where rents are usually higher. The gross rent is directly coded and easy to recover. In Table 2, one can observe that the average gross rent is about 650 euros while

**Table 2. Price, surface, expenditures and type of lease.**

|  | Count | Mean | Std | Min | 25% | 50% | 75% | Max |
|---|---|---|---|---|---|---|---|---|
| Gross Rent | 4370365 | 652.3 | 405.2 | 81.0 | 449.0 | 567.0 | 740.0 | 24000.0 |
| Surface | 4370365 | 55.7 | 31.1 | 6.0 | 33.8 | 50.0 | 70.0 | 497.0 |
| Gross Rent per square meter | 4370365 | 13.6 | 7.3 | 0.2 | 8.9 | 11.8 | 16.2 | 1846.2 |
| Time elapsed since publication (days) | 4370365 | 29.4 | 39.0 | 0.0 | 8.0 | 20.0 | 38.0 | 770.0 |
| Expenditures (%): Included | 4370365 | 72.5 | 44.7 | 0.0 | 0.0 | 100.0 | 100.0 | 100.0 |
| Expenditures (%): Not Included | 4370365 | 5.8 | 23.3 | 0.0 | 0.0 | 0.0 | 0.0 | 100.0 |
| Expenditures (%): Unknown | 4370365 | 21.7 | 41.2 | 0.0 | 0.0 | 0.0 | 0.0 | 100.0 |
| Amount Expenditures | 623373 | 61.8 | 60.1 | 0.0 | 30.0 | 48.0 | 80.0 | 3705.0 |
| Collective heating (%) | 4370365 | 3.6 | 18.5 | 0.0 | 0.0 | 0.0 | 0.0 | 100.0 |
| Hot water (%) | 4370365 | 0.2 | 4.2 | 0.0 | 0.0 | 0.0 | 0.0 | 100.0 |
| Trash collection (%) | 4370365 | 4.6 | 20.9 | 0.0 | 0.0 | 0.0 | 0.0 | 100.0 |
| Furnished (%): No | 4370365 | 75.8 | 42.8 | 0.0 | 100.0 | 100.0 | 100.0 | 100.0 |
| Furnished (%): Yes | 4370365 | 24.2 | 42.8 | 0.0 | 0.0 | 0.0 | 0.0 | 100.0 |

the rent per square meter is around 13 euros. Both websites also provide additional information specifying whether the rent displayed includes extra expenditures (e.g., waste collection, water, heating). About 70% of the rent displayed includes some kind of extra expenditures. Unfortunately, the share of the rent attributed to these is not directly coded and is recovered from the text using regular expressions. The algorithm identifies whether the word "charges" is in the text and recovers the amount in euro around this word that is inferior to the rent. About 30% of the ads indicates the amount of extra expenditures. The average estimated amount of extra expenditures on the subsample is around 61.8 euros which represents 9% of the average rent. Moreover, we also estimate the average and median amount of extra expenditures for all the flats using the average amount of extra expenditures for dwellings with similar characteristics (based on the type of dwellings and number of rooms) in the same department for observations where the extra expenditures could not be recovered. This also allows us to estimate an average net rent per square meter for all municipalities and strata.

Based on the text, it is also possible to infer which type of expenditures are included as collective heating or trash collection. Lastly, a second important information is the type of lease indicating whether furnitures are included in the lease or not. This variable is of particular importance as the minimal length of the lease is 1 year when the flat is furnished or 3 years if not furnished. Again, if this information appears in the code of the web page for the most recent period, it was not systematically filled in the first waves. Consequently it is also coded from regular expressions identified in the description. We extract both variables, the one created from the text and the one based on the variables provided by websites. About 24% of the flats are offered as furnished. The publication date collected can also be used to compute the time elapsed since the publication was last observed. About 50% of the ads have disappeared after 20 days.

*The type of units and the number of rooms.* Each website has a specific part of the webpage dedicated to the type of unit and the number of rooms. No treatment is thus required and these variables are taken directly from the collected variables.

As shown in Table 3, most of the units are flats as houses only represent about 16% of the sample. Units are of a relatively small size as their vast majority have one or two rooms while the average surface was about 56 square meter. These characteristics are typical of the French rental market which is dedicated to younger people with few children.

*Floors and other amenities.* From the description, it is also possible to identify in the description the floor and amenities in the building. As shown in Table 4, the floor is recovered for 40% of the ads while 14% of the ads announce the presence of an elevator. A total of 36% have a balcony or a kitchen with some equipment. Lastly, 46% offer the possibility of parking a car.

**Table 3. Type of units, number of rooms and surface.**

|  | Count | Mean | Std |
|---|---|---|---|
| House (%) | 4370365 | 15.4 | 36.1 |
| Rooms (%): 01 | 4370365 | 21.0 | 40.8 |
| Rooms (%): 02 | 4370365 | 32.6 | 46.9 |
| Rooms (%): 03 | 4370365 | 26.1 | 43.9 |
| Rooms (%): 04 | 4370365 | 12.4 | 33.0 |
| Rooms (%): 05 | 4370365 | 5.4 | 22.6 |
| Rooms (%): 6+ | 4370365 | 2.5 | 15.6 |

**Table 4. Floors and other amenities.**

|  | Count | Mean | Std |
|---|---|---|---|
| **Floor** |  |  |  |
| Floor (%): 0.0 | 4370365 | 8.8 | 28.3 |
| Floor (%): 1.0 | 4370365 | 10.8 | 31.0 |
| Floor (%): 2.0 | 4370365 | 7.9 | 27.0 |
| Floor (%): 3.0 | 4370365 | 4.3 | 20.3 |
| Floor (%): 4.0 | 4370365 | 2.1 | 14.4 |
| Floor (%): 5.0 | 4370365 | 1.1 | 10.2 |
| Floor (%): 6+ | 4370365 | 1.2 | 10.8 |
| Floor (%): Unknown floor | 4370365 | 63.9 | 48.0 |
| **Amenities** |  |  |  |
| Elevator (%) | 4370365 | 14.5 | 35.2 |
| Double glazing (%) | 4370365 | 9.6 | 29.5 |
| Kitchen with equipment (%) | 4370365 | 35.4 | 47.8 |
| Garage (%) | 4370365 | 45.4 | 49.8 |
| Garden (%) | 4370365 | 17.3 | 37.8 |
| Balcony (%) | 4370365 | 35.4 | 47.8 |

# External validation of the dataset

## The coverage of the database

To assess the coverage of the dataset coming from our collection process, we consider that housing units observed are a subsample of the exhaustive rental market which is documented in the French Census [27]. We develop a simple method inspired by the adjustment on margin method [28].

1. From the census of the year 2016, we create many strata combining the location of the occupied private rental units (municipality) and their number of rooms. For example, the first arrondissement of Paris is composed of five strata defined by the number of rooms (1,2,3,4,5 or more). Each strata contains some observations in the census noted $N^c$ which represents the number of rental units in the strata observed in the census of a given year. Robustness checks including also vacant units are performed and reported in S2 Fig in S1 Appendix.

2. In the second step we assign our scraped ad to each strata. The number of scraped ads in each strata is noted $n^s$

3. The coverage (i.e. number of ads for each unit) is simply defined as $\frac{n^s}{N^c}$

Thus we can measure the coverage of each type of goods in two dimensions: their location and the number of rooms. For example: we can determine how many flats an ad with two bedrooms in the 1st district of Paris will be represented.

We use two different subsamples of the census to create two alternate measures. First, $N^c$ is defined using all the rental units. Second, $N^c$ is defined using the rental units occupied for less than 5 years used to proxy the flow of rental units on the market over our period of study. Fig 2 represents the distribution (weighted by the number of units) of the coverage of our strata while Fig 3 maps it across the French territory. On average each strata had one ad per unit rented for less than five years and 0.66 ads per rental unit. In rural areas, where the number of tenants is very low. The coverage is usually very high. In these places, the ratio can be even

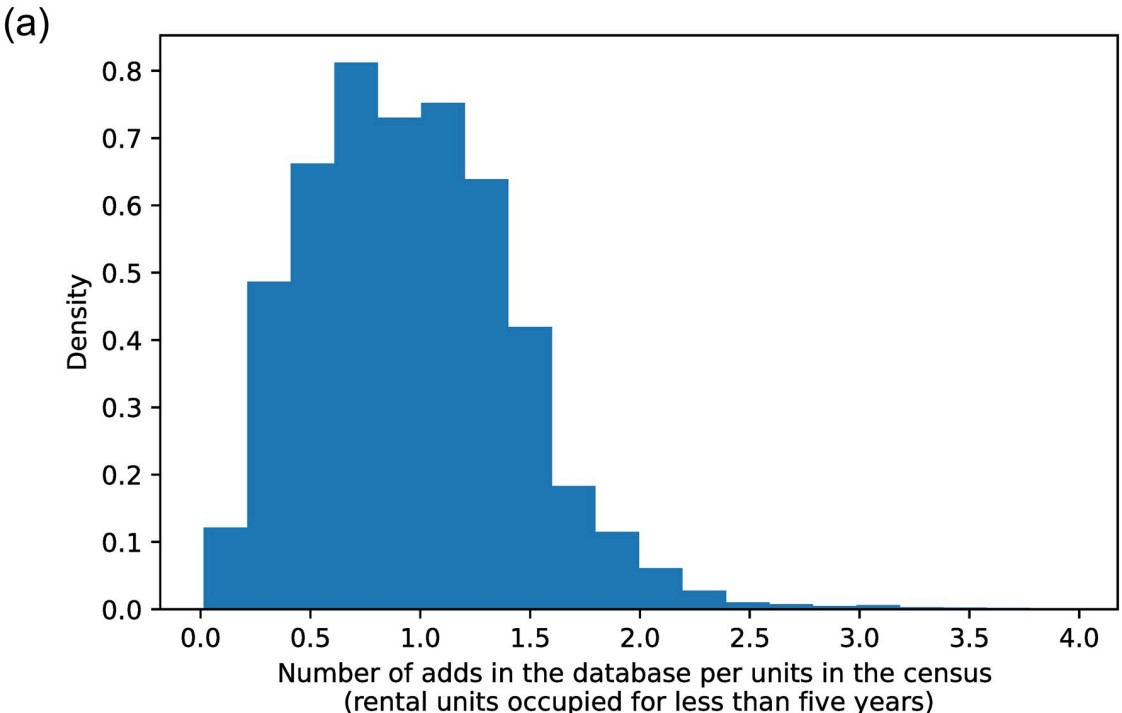

(a)

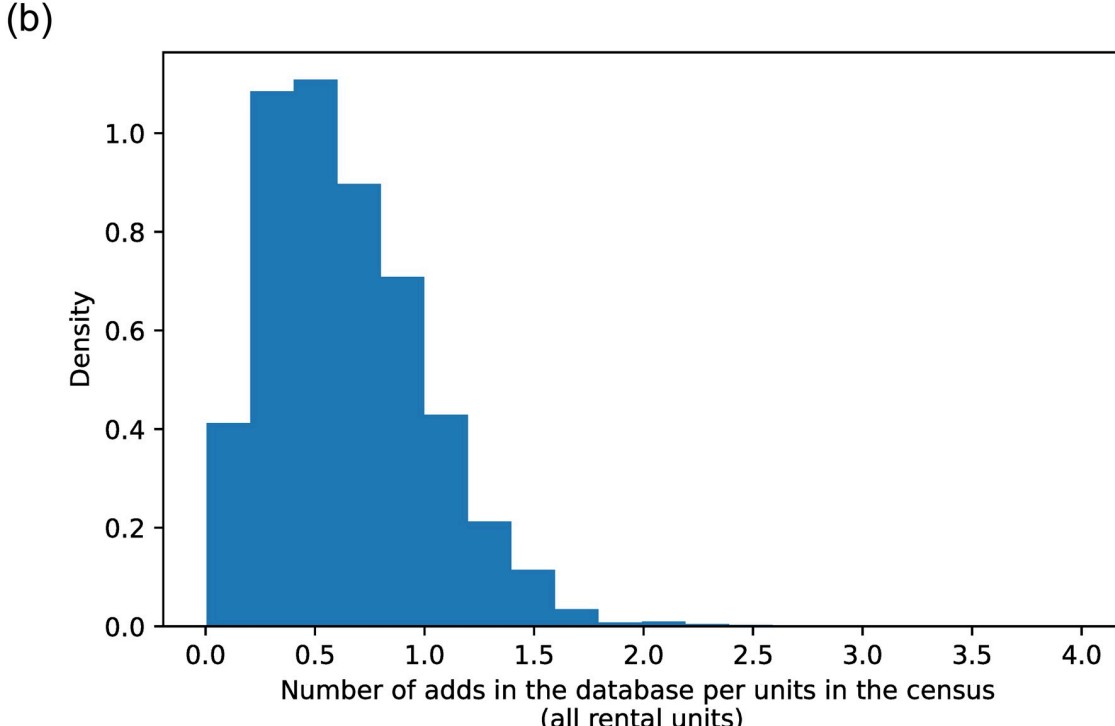

(b)

**Fig 2. Representativeness of the database.** *Note:* Distribution of the ratio between the number of ads scraped ($n^s$) and the number of rental units in the Census ($N^c$). Each observation is a strata weighted by its number of rental units ($N^c$). In panel a) $N^c$ is computed from rental dwellings occupied for less than 5 years. In panel b) $N^C$ is computed with all rental dwellings in the census.

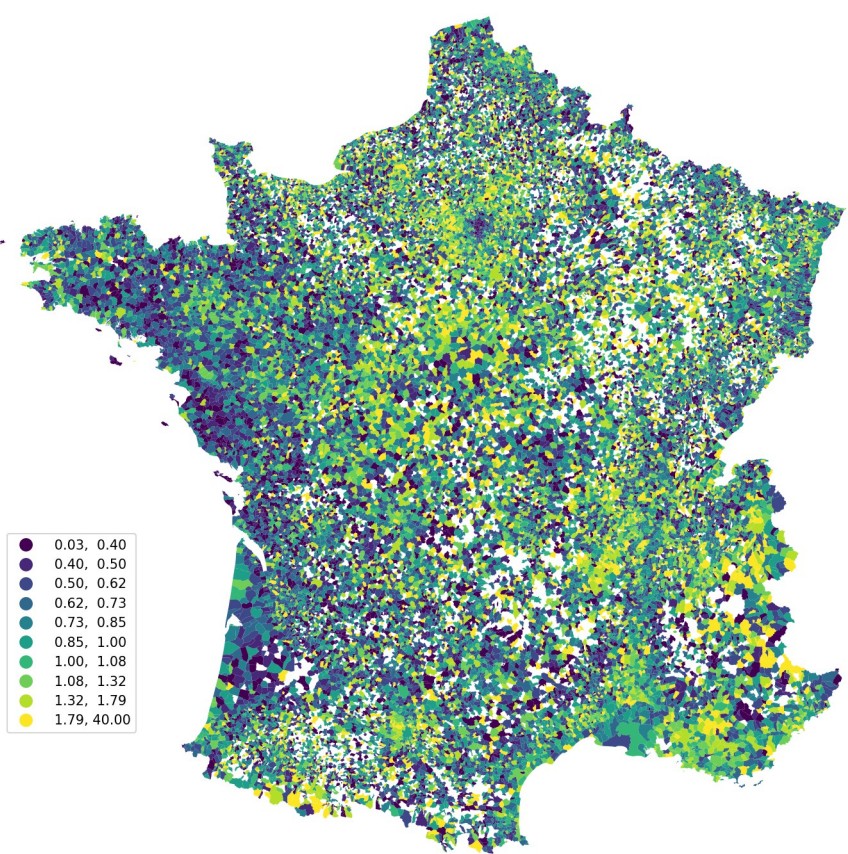

**Fig 3. Representativeness of the database through space.** *Note:* Average number of ads per unit in all strata of a municipality. Authors' computations and Admin Express Commune which is under an Open Licence Etalab https://www.data.gouv.fr/fr/datasets/admin-express/.

above one because there are very few rental units observed and thus one can observe more online ads than the number of tenants observed in the census. Moreover, some places can be vacant and for rents (see S2 Fig in S1 Appendix). Nevertheless, other tiny rural places are sometimes not covered by online ads. The coverage appears to be lower in city center where the number of tenants in the private sector is usually higher. For example, within Paris the average coverage among strata is around 0.5 ads per rental units occupied for less than five years. In these city, this coverage is expected to grow over time as we only have been scraping for 2 years. In some specific cities, the ratio can be also above one when the turnover is high and the same dwellings can be posted several time per year. This can be the case in areas where a large share of tenants are students or temporary workers in the tourism sector.

## Comparison with local rent observatories and CLAMEUR

If the type of housing unit observed and the channel used to find the flat appear fairly representative, posted rent might be different from the real one. Nevertheless several important observations lead us to believe that this bias remains limited. From a theoretical standpoint, if we model the housing market as a frictional market where a landlord and a tenant meet [29], the bargained rent is a weighted sum of the landlord's and the tenant's surpluses. The rent crucially depends on the relative bargaining power of the landlord/tenant. However, when the bargaining power of the tenant is close to zero the rent converges toward the posted rent when

we assume a price competition among landlords [30]. Moreover, the bargaining power in the Nash bargaining process can be seen as a factor of relative impatience where the impatient party has a lower bargaining power [31]. The lack of housing supply in France, particularly in large cities, suggests that prospecting people have a relatively small bargaining power at least in the major urban areas. Moreover, for other markets, we expect that the transparency of the online platforms where landlords can observe at a reduced cost the prices and movements of their competitors offering a similar unit in the same area can also drive the posted rent close to the market rent.

For these reasons posted rents are not likely to differ too much from the actual ones signed on the contract. Therefore, we compare our data with the statistics on rents published by Local Observatories (OLL) based on surveys.

OLL publishes yearly statistics on local rents based on regular surveys for 35 urban areas [32]. These datasets provide statistics on the first quartile, the median, the third quartile and the average net rents per square meter for several sampling areas within each urban areas. These statistics are computed for some groups of units with homogeneous characteristics (House vs apartments; number of rooms; period of completion of the building). Interestingly, while the survey conducted by OLL focuses on the whole rental sector, it also publishes statistics on new leases signed for less than one year. We thus recovered these statistics on new leases to compare them with the ads published in the preceding year. This allows us to test whether there exists some discrepancies between the net rents of online ads and the net rent actually paid by tenants and measured through surveys. The correlation between the median rent computed from both datasets is 0.95, as is the correlation coefficient between average rents from both sources. Fig 4 represents this correlation between median and the distribution of rents computed from both datasets with the first and the third quartiles. We observe that the distribution is concentrated around the 45 degree lines. Unfortunately neither OLL nor Clameur provides confidence interval for their estimates. Panels a) and b) from Table 5 tests whether there is a significant difference between the average and median rents computed from ads and the statistics published by the OLL estimating the following equation with ordinary least Squares:

$$Rent_{d,a,r} = \alpha + \beta \times 1_{d=Ads} + X_{a,r}\lambda \qquad (1)$$

where $Rent_{d,a,r}$ is the average or median rent estimated using dataset d, for dwellings located in the area, a, with r rooms. $\alpha$ is a constant while $1_{d=Ads}$ is a dummy indicating when the observation belongs to the dataset created thanks to online ads. $\beta$, reported in Table 5, compare the difference between the averages measured from online ads and those measured by OLL. $X_{a,r}$ are fixed effects. In columns 1 and 4, no fixed effects are included. This corresponds to a simple comparison of averages. In columns 2 and 5 separate fixed effects indicates the number of rooms and the area; in such a case the comparison is performed between goods controlling for the influence of the number of rooms and the location of the good. Lastly, in columns 3 and 6 there is one fixed effect per strata (number of rooms x area), which insures that the comparison is made within each strata. Correlation of the residuals within each dataset and spatial correlation is accounted for clustering the standard errors at the urban unit level and at the dataset level. On average, the median rent is 0.0134 (95% CI -0.30,0.33) cents per square meter higher when computed with online Ads while the average rent is 0.324 (95% CI -0.12,0.78) cents higher. One should note that this difference remains small (about 2% of the average rent in the sample) and is not statistically significant. The fact that the average computed from online ads is slightly higher might reflect that some limited negotiation can happen on the market but also that part of the online activity covers dwellings with higher rents when compared with dwellings found by word of mouth.

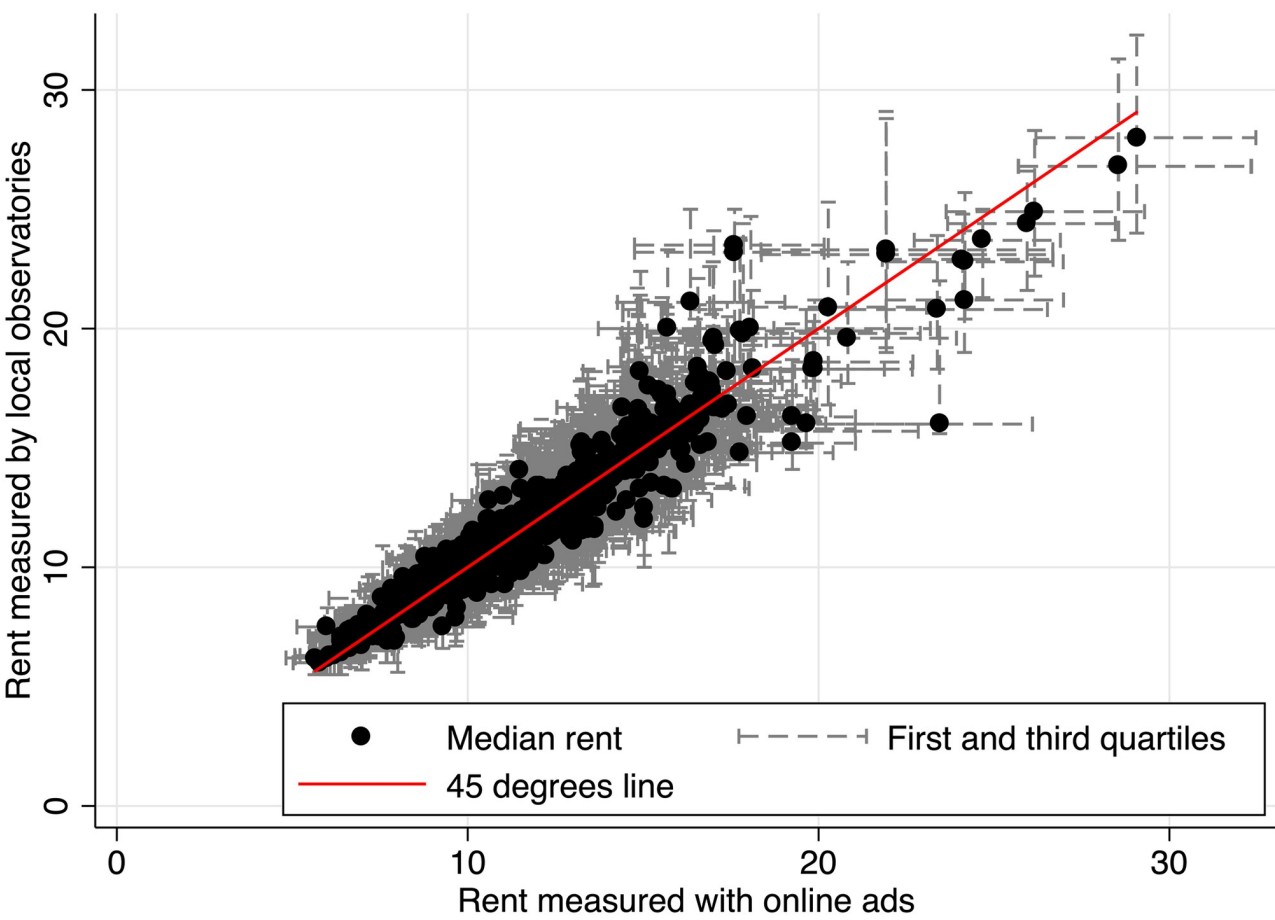

**Fig 4. Comparison of the rent distribution between OLL and ads, 2017.** *Note:* The correlation coefficient is 0.95, When estimating $Rent_{ads} = \beta Rent_{OLL}$ with OLS, the estimate of $\beta$ is 1.029 (95% CI 0.92 1.13) with a clustered standard error of 0.038.

CLAMEUR also publishes similar statistics on average rents by number of rooms for 887 municipalities. We thus also compare their statistics with these computed from online ads for similar dwellings. The correlation between the observations in Clameur and those produced by online ads is 0.94. Panel C) in Table 5 performs the same kind of exercise as the one realized for OLL. The results are comparable with these reported in panel A) and B). There is a small positive bias which is not statistically significant.

Notably, the absence of bias appeared true in tight housing markets with an inelastic housing supply but also in other market where competition also pushes landlords to reveal their reservation price. We illustrate this point in S1 Table in S1 Appendix, where the magnitude of the bias is similar in segments of the housing market above and below the median rent level. Our dataset thus appears consistent with alternate methods while providing a broader coverage of the territory with a moderate collection cost.

## Estimating hedonic models for local French housing markets

This new dataset on rents allows us to build hedonic models of the French local rental markets. These models can then be used to build a spatial constant quality hedonic rental price index for French municipalities to compare the cost of housing between territories. These can also be used to predict the rental value of dwellings from alternate data source. We pursue two

**Table 5. Comparison between the statistics provided by online ads and the OLL or Clameur.**

|  | (1) | (2) | (3) | (4) | (5) | (6) |
|---|---|---|---|---|---|---|
|  | Median Rent | | | Average Rent | | |
| Panel A) OLL—Appartments | | | | | | |
| Ads | 0.0134 | 0.0134 | 0.0134 | 0.324 | 0.324 | 0.324 |
|  | (0.0967) | (0.0998) | (0.0967) | (0.142) | (0.151) | (0.142) |
| N | 640 | 640 | 640 | 640 |  |  |
| R2 | 0.000 | 0.857 | 0.967 | 0.003 | 0.863 | 0.975 |
| Panel B) OLL—All types of dwellings | | | | | | |
| Ads | 0.0430 | 0.0430 | 0.0430 | 0.376 | 0.376 | 0.376 |
|  | (0.0754) | (0.0840) | (0.0754) | (0.153) | (0.162) | (0.153) |
| N | 738 | 738 | 738 | 738 | 738 | 738 |
| R2 | 0.000 | 0.843 | 0.973 | 0.003 | 0.849 | 0.980 |
| Panel C) Clameur—All types of dwellings | | | | | | |
| Ads | - | - | - | 0.471 | 0.471 | 0.471 |
|  | - | - | - | (1.125) | (0.179) | (0.194) |
| N | - | - | - | 7370 | 7370 | 7370 |
| R2 | - | - | - | 0.002 | 0.931 | 0.966 |
| Controls | | | | | | |
| N. rooms | N | Y | N | N | Y | N |
| Area FE | N | Y | N | N | Y | N |
| Area FE x N. rooms | N | N | Y | N | N | Y |

Standard errors in parentheses clustered at the Agglomeration levels and at the dataset level

$^*$ $p < 0.05$,

$^{**}$ $p < 0.01$,

$^{***}$ $p < 0.001$

*Note:* Estimates of $Rent_{d,a, r} = \alpha + \beta \times 1_{d = Ads} + X_{a,r}\lambda$. N. Rooms corresponds to the inclusion of rooms fixed effect (1,2,3,4+ in panel A and B or 1,2,3,4,5 in panel C). Area FE corresponds to the inclusion of area fixed effects. Sampling areas are defined by OLL and mostly cover groups of municipalities in panel A and B while these areas are 887 municipalities in panel C. Area FE x N. rooms corresponds to the inclusion of interaction terms between Area FE and N. rooms.

complementary applications with these hedonic models. First, we combine rental indices with equivalent indices based on prices to compute an hedonic rent-price ratio to document the discrepancies between rents and prices across the French territory. Second, we also use these models to predict the rental value of dwellings sold in 2016 and 2017 and the market value of subsidized dwellings belonging to the social sector where rents are administratively set and almost do not vary between municipalities. The latter allows us to estimate the in-kind benefit received by tenants living in the social housing sector and to compute the average subsidy received by social tenants for each French municipality.

## Hedonic models for rents and prices

We estimated an hedonic index of the rent and price for each French municipality where we had more than 10 observations:

$$ln(c_{i,s}) = ln(c^{ref}_{m(i)}) + X_i\beta_s + u_i \tag{2}$$

where $ln(c_{i,s})$ is the rent or price per square meter of unit i in strata s, $ln(c^{dref}_{m(i)})$ is the hedonic index of the municipality m where the unit is located, $X_i$ is a vector of hedonic characteristics

of the unit common to both datasets (surface, number of rooms, presence of other amenities (furnished, inclusion of extra expenditures)) and $\beta_s$ is the vector of corrective coefficients, which are allowed to vary between strata. We estimated our hedonic models separately for each year and each department for houses, apartments and a pooling of both. The specification and the use of logarithm as a dependent variable is standard in the literature on hedonic models and hedonic indices [4, 33, 34].

We estimate the rent index with our dataset removing also outliers below the 5th and above the 95th percentiles of rent per square meter following the standards in the literature [34]. Prices come from the administrative dataset of property transactions (Demande de Valeurs Foncières—DVF) which contains all real estate transactions in France (besides the Alsace-Moselle Region). It is available from 2014. This dataset has some advantages and drawbacks when compared with the datasets of French notaries [4]. Its main advantage is its exhaustiveness; thus, it represents all transactions that took place over the period that are representative from all French dwellings than what is observed on the rental market [3]. However, this comes at some cost: the number of hedonic characteristics is rather limited as it mainly contains the type of unit (apartment vs. House), the number of rooms, the surface, and the precise geocoding of the parcel. In estimating the rental value of social dwellings, the hedonic model as more variables as the social housing database (Répertoire des logements locatifs des bailleurs sociaux—RPLS) contrains more characteristics common with our dataset as the energy efficiency of dwellings and the floor. We also include neighborhoods fixed effects instead of municipality fixed effects when we have enough observations at the neighborhood level. This allows avoid relying on to contextual variables.

## The rent-price ratio in French municipalities

We compute the Rent Price Ratio for French municipalities using the following equation:

$$\frac{R}{P} = \frac{e^{\left( ln(r_{m(i)}^{ref}) + \frac{1}{2}\sigma_{r,s}^2 \right)}}{e^{\left( ln(p_{m(i)}^{ref}) + \frac{1}{2}\sigma_{p,s}^2 \right)}} \quad (3)$$

where $ln(r_{m(i)}^{ref})$ and $ln(p_{m(i)}^{ref})$ are respectively municipality levels hedonic indices for prices and rents, respectively, while $\sigma$ is the root-mean-square error of the corresponding regression used to remove the bias arising from the reverse transformation from logarithm [35]. In Fig 5, we plot the constant quality hedonic rent-price ratio index against the municipal hedonic price index. We observe that even when controlling for the characteristics of the dwellings, there remained some large spatial heterogeneity in the rent-price ratio, which is strongly correlated with the price levels of these municipalities. From this chart, we might infer that rents are not increasing as much as housing prices in tight markets. Panel A in Table 6 provides additional descriptive statistics on local rent, price, and rent-price hedonic indices for French municipalities. One also observes that the rent-price ratio is lower in municipalities belonging to large urban units or to the urban unit of Paris and for large municipalities with a population above 100,000 inhabitants. Similar discrepancies in the rent-price ratio have been put forward in other countries as in the US where superstar cities have a low rent-price ratio [36] or the UK [37]. The literature suggests that this might be explained by a higher expected growth rate of rents and the lower housing supply elasticity in the most expansive cities. Documenting, the main drivers of the spatial discrepancies is beyond the scope of this paper.

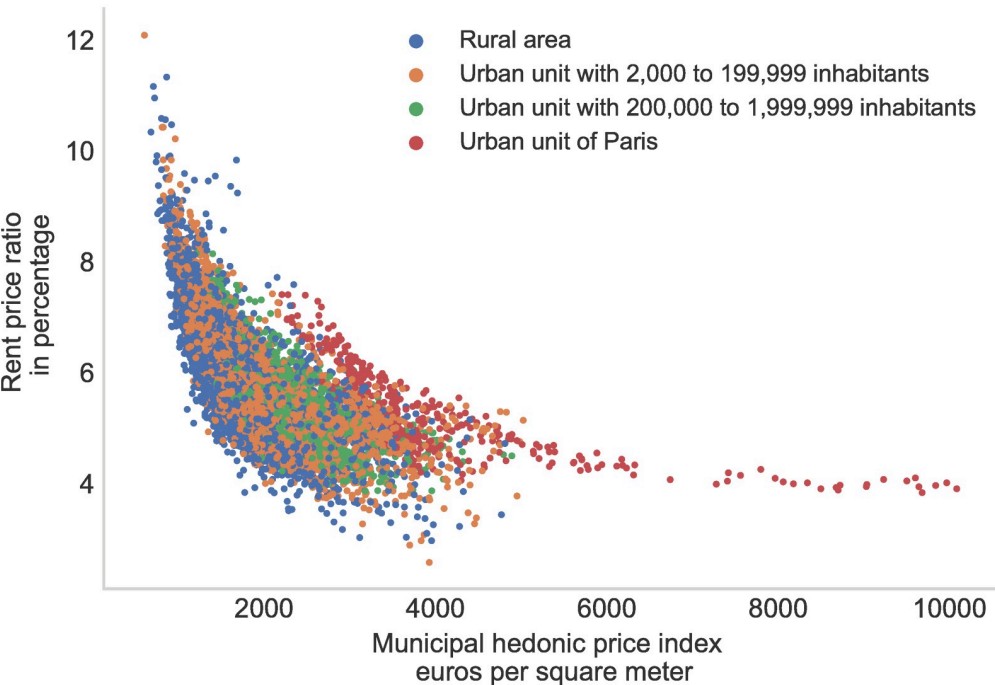

**Fig 5. Rent-price ratio in 2017.** *Note:* Price data are taken from the fiscal administrative database *Demande de Valeur Foncière*, rent data are taken from the database constructed by the authors. The rent-price ratio is calculated as the ratio of the municipal hedonic price index and the hedonic rental index. Only municipalities with more than 10 observations are displayed.

This strong spatial variation in the rent-price ratio demonstrates that the choice to measure the cost of housing due to prices or rents might affect the estimated elasticities between amenities and housing costs. For example, in a companion paper [38], we show that the cost of agglomeration—that is, the elasticity between density and housing cost in the city center [4]—lower when measured with rents than when measured with prices.

**Table 6. Descriptive statistics of the municipal hedonic indices.**

| | Monthly Rent *per square meter* | Yearly Price *per square meter* | Rent-Price Ratio *yearly in %* | Observations *municipalities* |
|---|---|---|---|---|
| Country level | 8.41 | 1748.00 | 6.14 | 22920 |
| Outside urban unit | 7.88 | 1597.00 | 6.29 | 16413 |
| Urban unit with 2k to 4k inhabitants | 8.67 | 1846.76 | 5.90 | 1642 |
| Urban unit with 5k to 10k inhabitants | 8.80 | 1892.00 | 5.83 | 1007 |
| Urban unit with 10k to 20k inhabitants | 9.05 | 1947.97 | 5.81 | 705 |
| Urban unit with 20k to 50k inhabitants | 9.00 | 1944.43 | 5.76 | 725 |
| Urban unit with 50k to 100k inhabitants | 9.24 | 1948.03 | 5.88 | 498 |
| Urban unit with 100k to 200k inhabitants | 10.37 | 2220.27 | 5.89 | 364 |
| Urban unit with 200k to 2,000k inhabitants | 10.56 | 2356.41 | 5.55 | 1137 |
| Urban unit of Paris | 16.41 | 3903.05 | 5.24 | 429 |

*Note:* The first panel provides estimates of rent and prices for individual dwellings of the *Demande de Valeur Foncière* administrative database. The second and third panels provide rent estimates and hedonic prices at the municipal level. Rents in column 3 and individual prices in column 4 are the hedonic prices and rents estimated with dwellings individual characteristics. Rent-price ratios are the ratio of the estimated prices and rents of columns 3 and 4. Municipal rents and prices were estimated from the fixed effects of hedonic models.

## Predicting the rent of real estate transactions and social dwellings

After the estimation of Eq 2, it is also possible to predict a rental value, a price, and thus a rent-price ratio for every transactions that we observe in DVF. The rent per square meter is simply obtained by substituting dwellings' characteristics in the estimated hedonic equation:

$$\widehat{R} = e^{\widehat{ln(r^{ref}_{m(i)})} + X_i\widehat{\beta_s} + \frac{1}{2}\widehat{\sigma}^2_{p,s}} \tag{4}$$

The descriptive statistics of this output are reported in Table 7. There are discrepancies in the rent-price ratio following the size of the dwellings. On average, the estimated rents of dwellings for sale in 2017 were 11,66 euros per square meter and were lower for houses than for apartments. The rent-price ratios also present clear discrepancies following the size of the dwellings. The average predicted rent-price ratio is 5.69 percent and ranges from 6.93% for small dwellings with one single room to 5.26% for large units with five rooms or more. These discrepancies remain even when investigating the patterns within municipalities as illustrated in S4 Table in S1 Appendix.

Eq 2 can also be used to predict the rental values of social dwellings. Social housing units often called "Habitations à Loyers Modérés" (HLM) represent about half of the rental market and thus between 14% ad 20% of the total housing stock in France. They are mostly owned and managed by non-profit and public landlords. Contrary to the private sector, their rent is controlled and set administratively, as a rent ceiling per square meter is imposed to social landlords. Moreover, households under an income ceiling have to be registered on a waiting list to access these units which are assigned by a commission. One interesting feature of the sector comes from the fact that all rents have been systematically registered in a centralized file, the RPLS, since 2011. To account for the spatial concentration of social dwellings, we substitute municipal fixed effects with neighborhood (sections cadastrales) fixed effects. It is worth noting that these neighborhood fixed effects allow us to avoid relying on contextual variables (average income, socioeconomic composition of the neighborhood, etc.).

The results are reported in Table 8. The implicit subsidy is obtained substrating the rent paid by social tenants from the estimated market rent. In relative terms, the implicit subsidy represents about 46.6% of the rental value of the unit. In absolute terms, the average subsidy is between 370 and 390 euros while the median was around 300 euros. The average subsidy is 110 euros larger than in a previous study [39]; three reasons might be invoked to explain such

**Table 7. Descriptive statistics on the transaction-level rent-price ratio.**

|                   | Monthly Rent *per square meter* | Yearly Price *per square meter* | Rent-Price Ratio *yearly in %* | Observations *dwellings* |
|-------------------|---------------------------------|---------------------------------|--------------------------------|--------------------------|
| All               | 11.66                           | 2641.79                         | 5.69                           | 714437                   |
| Rooms: 1          | 18.85                           | 3715.98                         | 6.93                           | 57434                    |
| Rooms: 2          | 14.90                           | 3288.53                         | 6.11                           | 113932                   |
| Rooms: 3          | 11.73                           | 2685.55                         | 5.74                           | 172100                   |
| Rooms: 4          | 10.01                           | 2349.93                         | 5.43                           | 190975                   |
| Rooms: 5+         | 9.01                            | 2157.48                         | 5.26                           | 179996                   |
| Type: apartment   | 15.14                           | 3379.90                         | 5.95                           | 312935                   |
| Type: house       | 8.95                            | 2066.50                         | 5.49                           | 401502                   |

*Note:* The first panel provides estimates of rent and prices for individual dwellings of the *Demande de Valeur Foncière* administrative database. The second and third panels provide rent estimates and hedonic prices at the municipal level. Rents in column 3 and individual prices in column 4 are the hedonic prices and rents estimated with dwellings individual characteristics. Rent-price ratios are the ratio of the estimated prices and rents of columns 3 and 4. Municipal rents and prices were estimated from the fixed effects of the hedonic models.

Table 8. Descriptive statistics on the implicit subsidy of social housing.

| | Total | | | Per square meter | | | N |
|---|---|---|---|---|---|---|---|
| | Market Rent | Subsidized Rent | Subsisdy | Market Rent | Subsidized Rent | Subsisdy | |
| Panel A) Characteristics of the dwellings | | | | | | | |
| All dwellings | 771.52 | 386.93 | 390.04 | 12.10 | 5.96 | 6.19 | 4786527 |
| 1 | 536.43 | 247.75 | 297.68 | 17.76 | 8.03 | 9.72 | 254433 |
| 2 | 646.60 | 316.63 | 331.75 | 13.57 | 6.56 | 7.03 | 942511 |
| 3 | 752.51 | 378.08 | 378.41 | 11.79 | 5.85 | 5.99 | 1793861 |
| 4 | 854.29 | 435.61 | 426.78 | 10.91 | 5.51 | 5.49 | 1397673 |
| 5+ | 1012.71 | 511.26 | 511.82 | 10.58 | 5.31 | 5.39 | 398049 |
| Panel B) Type of subsidy received | | | | | | | |
| PLAI | 730.90 | 360.97 | 373.98 | 11.99 | 5.69 | 6.25 | 257518 |
| PLI | 985.37 | 575.49 | 440.64 | 15.60 | 8.82 | 7.22 | 145522 |
| PLS | 911.03 | 544.31 | 376.96 | 14.80 | 8.61 | 6.13 | 251870 |
| PLUS | 758.02 | 372.99 | 390.21 | 11.82 | 5.72 | 6.16 | 4131617 |
| Panel C) Area | | | | | | | |
| 01 | 1028.85 | 420.11 | 611.69 | 16.41 | 6.66 | 9.72 | 840792 |
| 01 bis | 1407.07 | 453.83 | 959.54 | 24.32 | 7.82 | 16.46 | 409041 |
| 02 | 691.22 | 382.28 | 312.07 | 10.60 | 5.80 | 4.81 | 2063051 |
| 03 | 560.71 | 355.06 | 208.40 | 8.35 | 5.23 | 3.15 | 1473643 |

*Note:* The table presents the observed and estimated rents for the RPLS. Column (1) shows the estimated average rent for the private market, column (2) the observed average rent in the social stock, column (3) the average social benefit. Columns (4) to (6) present the same variables at the square meter level.

a result. First, rents have been increasing since 2006, and thus the subsidy should be at least around 320 euros by the sole effect of inflation of both rents, a level close to our median estimates. Second, previous estimates were based on a former leases (in stock), while we estimate on new leases. Third, and more importantly, previous estimates were based on a survey that was probably not representative of the whole distribution of rents in the social housing sector. This is a concern for Paris, which represents about 5% of social dwellings. The implicit subsidy for the French capital appears extremely high for a large number of units and increases dramatically the average. Indeed, the right tails of the distribution of the subsidy were exclusively composed of housing units located in Paris, where the subsidy can be well above 1,000 euros. Typically, a social housing unit in the center of Paris with a surface above 80 square meter and a controlled rent below 600 euros while its market rent is estimated to be above 2,500 euros. The absence of these units in the French housing survey might have biased the average subsidy in previous contributions. This is easily perceptible by the large discrepancy (80 euros) between the average and the median subsidies in our results. By excluding Paris from our dataset, the average subsidy declined to 335 euros. Lastly, by excluding Paris urban area, the average subsidy dropped to 255 euros. However, it is worth noting that the RPLS does not contains any information about the complementary rents (supplément de loyer solidarité) that the wealthiest households should pay when occupying some social housing units in desirable areas. This might decrease rent savings for some desirable units, even if many administrative reports estimate that these additional rents are only mildly applied. In a nutshell, our estimates are in line with previous studies, while gaining access to the whole distribution of social housing units shows that the implicit subsidy might be more unevenly distributed.

If there exists different categories of social dwellings adjusting the maximum rent following household's income and the location of the apartment, these parameters play a very limited

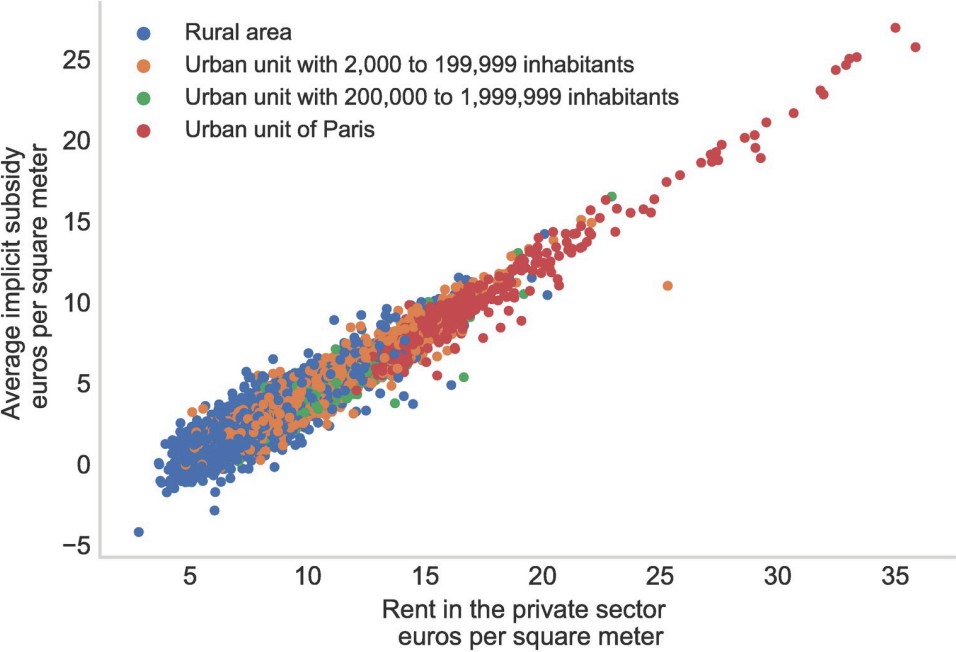

**Fig 6. Average subsidy in the main French municipalities.** *Note:* This figure provides relationship between the estimated average rent in the private sector per municipality (x-axis) and the implicit housing subsidy per municipality (y-axis). Characteristics of the dwellings are those of the RPLS, private rents are estimated from our data.

role in explaining the magnitude of the implicit subsidy, which is almost exclusively driven by the location of the dwelling as illustrated in Fig 6. For example, while the average private rent per square meter in the center of Paris is 35 euros, the rent cap remains around 6.09 euros for the most common type of social dwellings. In contrast, private rents can be less than half in the Paris suburbs, while the rent ceiling falls to 4.7 in the social housing sector. In other words, the rent gradient of the urban area for social dwellings is almost flat while it steeply declines in the private sector, resulting in a very large relative implicit subsidy in the center of Paris [40]. The importance of location in the implicit subsidy is also illustrated in Table 8 by the fact that PLI, which are social dwellings designed for the upper-middle class, offer a higher amount of implicit subsidy than standard social dwellings because they tend to be concentrated in more attractive municipalities.

S5 Table in S1 Appendix confirms this point in a more systematic way: regressing the estimated subsidy on dwelling's characteristics only explains a limited share of the variance while including geographical fixed effects as social housing areas or municipality fixed effects increase dramatically the R2. S6 Table in S1 Appendix also shows that, net of dwellings' characteristics, the city level of implicit subsidy is mostly explained by the level of private rent.

This uneven spatial distribution of the implicit subsidy has a direct consequence on the redistributive profile of the policy. Indeed, as richer households tend to remain for a longer time in good-quality social dwellings thanks to the right of security of tenure [41], they tend to be concentrated in social dwellings located in wealthier municipalities thus benefiting from the largest in kind subsidy as illustrated in Fig 7. Columns 2 and 3 in S6 Table in S1 Appendix show that this correlation holds controlling for the share of vacant dwellings and the size of the municipality. It is noteworthy that the inclusion of urban area fixed effect in column 4 reduces this correlation indicating that richer social tenants tend to be located in larger urban units as

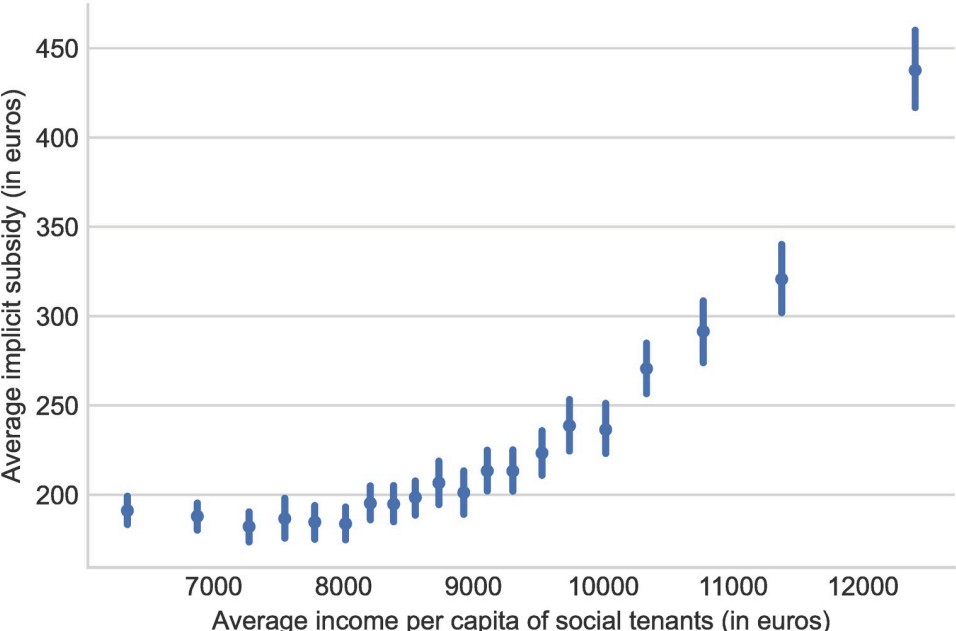

**Fig 7. Average municipal social housing implicit subsidy and income per capita of social tenants.** *Note:* This figure displays the relationship between the average per capita income of social housing tenants per municipality in 2013 (x-axis) and the average level of implicit subsidy of social housing per municipality (y-axis). Municipalities have been grouped by quantiles of per capita income of social housing tenants. Characteristics of the dwellings are those of the RPLS, private rents are estimated from our data.

Paris or Lyon. As a consequence, the correlation between income and subsidy is smaller within agglomerations. Overall, our results are in line with previous findings in Finland [42] where it has been shown that social housing are less redistributive than housing allowances as they are harder to concentrate on poorest households.

## Conclusion

In this paper, we describe a new data collection technique to provide accurate data on local housing markets for researchers and statisticians using online data. As we attempt to demonstrate, this can provide a relatively cheap and precise way to collect an important amount of micro data to answer research questions related to market dynamics. If these online data correspond to posted rents and not to signed contracts, the relative transparency of online platforms may force landlords to reveal the market price. The comparison between our dataset and standard surveys supports this intuition. Indeed, no significant difference is observed between the average rent computed from online ads and from local rental observatories, which strengthens our confidence in the reliability of research papers based on these types of data.

We also present systematic evidence of the divergence between measures of housing costs based on prices or rents documenting the spatial variation of the rent-price ratio in France. This stresses the fact that the capitalization of amenities might be different according to the type of data used.

Finally, we also use these data to calibrate hedonic models in order to predict the rental value of transactions and social dwellings, which we provide as an additional output of our paper. The estimated rent for social dwellings allows us to document the spatial disparities in the in-kind subsidy received by social tenants in France, which tends to be positively correlated

with the income of these households, confirming more systematically previous findings based on local [39, 41] or international [42] studies.

To conclude, we posit that these data offer many potential applications and should allow to tackle new research questions. For example, these data can be used to assess the rental value of the housing stock for research or taxation purposes [3], for public policy evaluation [18], or to assess the impact of the development of Airbnb on rental markets [19]. It can also be used to test spatial equilibrium models [43]. Finally, such a technique offers possibilities in countries where the collection of data is difficult and costly, as in developing economies [44].

## Supporting information

**S1 Appendix.**
(PDF)

**S1 File.**
(ZIP)

## Acknowledgments

The authors also thank participants in the Large Open/Online Raw Dataset (LOORD) and Numimmo seminars for their comments and questions. They also thank the editor Nils Kok and two anonymous referees for their helpful comments. They are particularly grateful to Antonin Bergeaud, Jean-Charles Bricongne, Julia Cagé, Gabrielle Fack, Gilles Duranton, Laurent Gobillon, Morgane Laouennan, Philippe Martin, Joan Monras, Florian Oswald, Bruno Palier, Quentin Ramond, Marco Schmid, Claude Taffin, Corentin Trévien, Grégory Verdugo, Paul Vertier, Benjamin Vignolles and Etienne Wasmer for their helpful comments and discussions.

## Author Contributions

**Writing – original draft:** Guillaume Chapelle, Jean Benoît Eyméoud.

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
