## [Decision Letter · Decision Letter 0]

18 Mar 2021

PONE-D-21-03045

Can big data increase our knowledge of local rental markets?  A dataset on the rental sector in France

PLOS ONE

Dear Dr. Chapelle,

I have now received the reports from two expert referees, and I have your manuscript myself. One referee is of the opinion that the manuscript should be rejected, while the other referee suggests revisions that are quite doable. My own opinion is in the middle -- the current version of the paper is not interesting enough for PLOS One, but, with hard work I can see a version of this paper that is more suitable. Most importantly, the paper shouldn't just focus on the novelty of web scraping and your ability to build a database. You should actually do something with the data, e.g. build an index, provide an interesting analysis that would otherwise be impossible in France, etc. I see this as a challenging revision and there is no guarantee that, should you resubmit a revised version of the paper, the revision will be accepted. I would fully understand if you don't pursue a publication in PLOS One and send the paper to a different journal.

If you decide to move forward, please submit your revised manuscript by Apr 24 2021 11:59PM. If you will need more time than this to complete your revisions, please reply to this message or contact the journal office at plosone@plos.org. Please include the following items when submitting your revised manuscript:

We look forward to receiving your revised manuscript.

Kind regards,

Nils Kok

Academic Editor

PLOS ONE

Journal Requirements:

"The authors acknowledge the support from ANR-11-LABX-0091 (LIEPP) and

ANR-11-IDEX-0005-02. They also thank participants in the Large Open/Online Raw

Dataset (LOORD) and Numimmo seminars for their comments and questions. They are

particularly grateful to Jean-Charles Bricongne, Julia Cage, Gilles Duranton, Laurent

Gobillon, Morgane Laouennan, Philippe Martin, Joan Monras, Florian Oswald, Bruno

Palier, Quentin Ramond, Marco Schmid, Claude Taffin, Corentin Trevien, Gregory

Verdugo, Paul Vertier, Benjamin Vignolles and Etienne Wasmer for their helpful

comments and discussions."

We note that one or more of the authors are employed by a commercial company: Banque de France.

(2) Please also provide an updated Competing Interests Statement declaring this commercial affiliation along with any other relevant declarations relating to employment, consultancy, patents, products in development, or marketed products, etc.  

5. Please ensure that you refer to Figures 1 and 4 in your text as, if accepted, production will need this reference to link the reader to the figure.

6. We note you have included a table to which you do not refer in the text of your manuscript. Please ensure that you refer to Table 2 in your text; if accepted, production will need this reference to link the reader to the Table.

7. We note that Figures 2 and 4 in your submission contain map images which may be copyrighted. All PLOS content is published under the Creative Commons Attribution License (CC BY 4.0), which means that the manuscript, images, and Supporting Information files will be freely available online, and any third party is permitted to access, download, copy, distribute, and use these materials in any way, even commercially, with proper attribution. For these reasons, we cannot publish previously copyrighted maps or satellite images created using proprietary data, such as Google software (Google Maps, Street View, and Earth). For more information, see our copyright guidelines: http://journals.plos.org/plosone/s/licenses-and-copyright.

(1) You may seek permission from the original copyright holder of Figures 2 and 4 to publish the content specifically under the CC BY 4.0 license. 

Reviewers' comments:

Reviewer's Responses to Questions

**Comments to the Author**

1. Is the manuscript technically sound, and do the data support the conclusions?

Reviewer #1: Partly

Reviewer #2: Yes

2. Has the statistical analysis been performed appropriately and rigorously? 

Reviewer #1: Yes

Reviewer #2: Yes

3. Have the authors made all data underlying the findings in their manuscript fully available?

Reviewer #1: No

Reviewer #2: No

4. Is the manuscript presented in an intelligible fashion and written in standard English?

Reviewer #1: No

Reviewer #2: Yes

5. Review Comments to the Author

Reviewer #1: For more details, please see the attachment

Overall, the authors did a great job creating this extensive dataset. How- ever, even though the presented methodology is advertised as low-cost, poten- tial obstacles and the required effort should be better discussed. The authors had to scrape the websites on a monthly base for around 2 years (l. 105) and used cooperative websites, providing public APIs (l.100). This effort might not be suitable for some purposes, such as when needing timely data or his- torical data. Furthermore, many similar websites in other countries are less cooperative, employing software to actively prevent scraping.1. However, my main problem with the study is the lack of originality in its current form. As explained below, there are commercial software solutions for web-scrapping technology. Furthermore, other studies have already explored scrapped real estate data or use them actively to answer other research questions. The authors discuss potential applications using “better”, scrapped data. Maybe some of these could be further explored to set the study apart from similar studies.

Reviewer #2: The purpose of this short paper is to present a novel database of housing rents in France. As is the case in many other countries, there exist few easily accessible databases of rental prices. The authors scrap data from two major real estate websites to obtain a large database of 4.3 million housing rents in France, covering the period from December 2015 until January 2018. The authors provide descriptive statistics and examine the representativeness of their database by comparing it to other databases.

From my reading of the paper, the data collection seems to have been executed properly and the resulting database could help us to acquire some new knowledge on the functioning of rental markets, in particular in France.

I provide my main comments and suggestions to this study below.

Major comments:

1. From the manuscript it does not become clear where the data will be published. The manuscript replicates the sample text “ALL XXX files are available in the Open Science Framework Repository (accession number(s) XXX, XXX.)”, but did not replace the XXX with the right information. At the OSF Repository, I was unable to find the database using the paper title. It is also not explicitly indicated that data will only be published at a later stage.

2. There are quite a few typos and grammatical errors in the manuscript, and I would suggest having the manuscript read and verified by a native copy editor. I am not a native speaker myself, but at the bottom of this review, I have provided a few examples from the first page. I have ignored them later in the manuscript because this would make my review lengthy.

Minor comments:

1. In the opening lines of the paper, the authors argue that historically the French authorities have recorded housing transactions and expressed limited interest in recording rental prices. I do not fully agree with this statement. For extended periods of time (at least until the mid-20th century) the French fiscal administration has been keeping enormous registrations of rental contracts, for example in the Enregistrement, since many taxes were based on rental prices rather than sales prices.

2. In the tables, the Min / 25% / 50% / 75% / Max values generally do not seem to add much, since most of the variables that are presented are dummy variables rather than continuous variables. I would suggest removing these statistics.

3. It would be useful to briefly mention the actual websites and companies that were used to collect the data and to provide some statistics on their market share / user base.

4. I would suggest adding some references to comparable work in other countries. Most notably, Boeing & Waddell (2016, Journal of Planning Education and Research) have scraped data from Craigslist in a very similar fashion.

5. For the representativeness of the database, it seems important to also consider the presence of social housing (HLM and other types), which are likely reported in the census but, based on what I assume, will not be published on these online websites.

6. The section about the comparison to the French census is at times confusing. For example, in line 190: “In a second step we assign our posted scraped to each strata”. It is not clear to me what “posted” is (it might be a typo). In line 193 it is unclear what “goods” refers to.

Some example typos from the first page:

Line 26: as the transaction is taxed instead of “when”

Line 36: “collecting information paid by …”: should “information” be “rental prices”?

Line 47: “this data sets … limits”: should be “these” and “limitations”

Line 52: “from insurance. Its provides ..”: should be “insurance companies” and “It provides”

Line 55: “on local market” should be “on local market conditions” or “on local markets”

6. PLOS authors have the option to publish the peer review history of their article (what does this mean?). If published, this will include your full peer review and any attached files.

Reviewer #1: No

Reviewer #2: No

---

## [Decision Letter · Decision Letter 1]

22 Jul 2021

PONE-D-21-03045R1

Can big data increase our knowledge of local rental markets?  A dataset on the rental sector in France

PLOS ONE

Dear Dr. Chapelle,

Thank you for submitting your manuscript to PLOS ONE. I have now received two referee reports on your paper submission. One referee recommends accepting the paper, while the second referee has gone from advising to "reject" to advising a "major revision". Looking at the comments of the referee, I'm somewhat more optimistic and would recommend a "minor revision." Much, if not all, of the feedback can be incorporated quite easily. To speed up the process, I'll likely not go back to the referee, but please provide a detailed response to each of the comments.

We look forward to receiving your revised manuscript.

Kind regards,

Nils Kok

Academic Editor

PLOS ONE

Journal Requirements:

Reviewers' comments:

Reviewer's Responses to Questions

**Comments to the Author**

1. If the authors have adequately addressed your comments raised in a previous round of review and you feel that this manuscript is now acceptable for publication, you may indicate that here to bypass the “Comments to the Author” section, enter your conflict of interest statement in the “Confidential to Editor” section, and submit your "Accept" recommendation.

Reviewer #1: (No Response)

Reviewer #2: All comments have been addressed

2. Is the manuscript technically sound, and do the data support the conclusions?

Reviewer #1: No

Reviewer #2: Yes

3. Has the statistical analysis been performed appropriately and rigorously? 

Reviewer #1: I Don't Know

Reviewer #2: Yes

4. Have the authors made all data underlying the findings in their manuscript fully available?

Reviewer #1: No

Reviewer #2: Yes

5. Is the manuscript presented in an intelligible fashion and written in standard English?

Reviewer #1: Yes

Reviewer #2: Yes

6. Review Comments to the Author

Reviewer #1: Summary

The authors present a new rent dataset for France, which was generated using web scrapping of online ads. Motivated by a lack of good French rental data, the growing coverage of real estate websites provides an opportunity for researchers to use data from online ads (l.31 – l.103). The authors describe how they used web scrapping over period of time to generate the underlying data and present some descriptive statistics (l.105 – l.201). Compared to “traditional” datasets of smaller size or inferior quality (less coverage or fewer information), the new dataset proofs to be a reliable source (l.202 – l.300). To demonstrate the potential of the new data, the authors estimate local hedonic indexes (l. 301 – l.340) and use these to estimate rent-to-price ratios (l.341 – l.364) and estimate free market rents for social housing properties to calculate the implied housing subsidies (l.365 – l.455).

Feedback

Overall, the authors did a great job creating this extensive dataset for France, showing its validity and using it for two potential applications. However, as explained in more detail below, I have some concerns about the overall validity of the study. The data collection process was performed in real-time (monthly) and through website APIs, meaning it cannot be done retroactively or might work in other countries. The described methodology is therefore not easily replicable but requires (extensive) local adaption on a case-by-case base, making the contribution rather descriptive.

One contribution is certainly the validation of online ads as a data source, being unbiased and in line with other datasets. However, other studies mentioned by the authors already show similar validation, even for France.

A valuable extension is certainly the application part, using the newly online data for specific use cases. However, this part requires way more attention, from a more extensive motivation (why are rent-to-price ratios at low aggregation good to have in France), over references for the utilized models, to the implications of using better data.

I would personally probably increase the application part and decrease the descriptive part.

Major Issues

• l.208: Nc needs more explanation (e.g. at what frequency is it collected). In line with the Notes of Figure 2, I understand ns is the number of online ads per strata (e.g. properties in the market) and Nc is the number of units available. Does this mean all properties (including occupied) or only in the market (and if so, over which frequency)? In the former case, a ratio of 1 would mean the online data contains as many units in the market as available or put differently, it means all apartments in the strata are on the market. Let’s assume the latter case. In this case I wonder how to interpret a ratio higher than 1? Does it mean there is more than one online add per available unit? In this case, I question the duplication filter. The authors need to be more specific.

Based on the sentence in line 225, the overall exercise seems not like a measure of representativeness but turnover, as it is suggested the ratio increases over time. I therefore understand Nc is a local constant while ns is time dependent. It would be great to have a reference point from the literature as 1 seem a very high value (meaning every apartment is on average sold once within 2 years). Overall, the whole part is just very confusing raising strong doubts about it. In Figure 3 the ratio even goes up to 40 which I cannot explain. If you have any references for this methodology, I strongly advise to use them here and be more specific about the whole test.

• The data should have been truncated using rent per square meter directly, not by price and square meter separately. As a result, there are still outliers in the data (e.g., minimum rent per square meter: 0.2 Euro in Table 2). Please investigate.

• l.323: I am confused why the authors call it an index but estimate the model for each year individually (maybe this is just a wording problem)? Also, I don’t understand how ln(c ref) is estimated. Technically, I understand that the local constant is seen as the index here, which would be in line with the literature. However, how is the logarithm applied or why is it assumed that the estimated constant is actually the logarithm. Please provide more details on the model derivation or provide some references to studies using a similar estimation. Interestingly, the authors later retransform the logarithms. Why not using levels directly then? Also, at which point is the calculation adjusted for property size or is the estimation on a per-square meter base? I think this section requires some rework and more explanations.

• There are no units in Table 6. E.g. taking Paris as an example, I don’t understand what 16.41 is. If this is in Euro, I assume it is per square meter per month, which would contradict the average in l. 427 though. This would raise the questions about the timely difference between rents and prices (rents would need to be adjusted for year or is this the monthly rent-price ratio?). In Figure 5, which is not linked to the text, it is indicated that the unit is percent.

•

Minor Issues

• The used tense is constantly changing (present, past, etc.)

• L.190 This is confusing. The authors say that they use public APIs to get the data but then state that they use HTML code to receive information. I am not aware of any API that provides HTML code. Does this mean the authors used the API and scrapped the website (double work so to say). In this case I wonder how much data could be generate by one or the other.

• Table 1: It seems like online ads are not representing the full spectrum of the rental market. Social housing seems to be excluded. Does this have implications on the estimations of free market rents for social units. Curious to hear the authors opinion.

• L.207 This is unclear and I am not sure if “crossing“ is the right word here.

• L. 221 vs. l.222 these sentences somehow contradict each other, maybe clarify.

• Please provide more information on the Strata (e.g., number of inhabitants)

• Table 3: What is a single unit or more specific what is the difference to 1 room?

• Equation (1) what is the unit of “a”? If it is Strata, I don’t understand how area fixed effects can be used (perfect collinearity)? Please elaborate a bit more and present the degrees of freedom.

• Figure 4: It also seems that the relationship is not fully 45 degrees as it diverges for higher priced areas (ca. above 20 median rent), indicating that online ads are higher for these areas (observations).

• l.362 the referenced paper is not by the same authors, so I would change the wording as readers might want to check the companion study.

• l. 470 please elaborate on this conclusion

Reviewer #2: (No Response)

7. PLOS authors have the option to publish the peer review history of their article (what does this mean?). If published, this will include your full peer review and any attached files.

Reviewer #1: No

Reviewer #2: No

---

## [Editor Report · Decision Letter 2]

10 Nov 2021

Can big data increase our knowledge of local rental markets?  A dataset on the rental sector in France

PONE-D-21-03045R2

Dear Dr. Chapelle,

We’re pleased to inform you that your manuscript has been judged scientifically suitable for publication and will be formally accepted for publication once it meets all outstanding technical requirements.

Kind regards,

Nils Kok

Academic Editor

PLOS ONE

Additional Editor Comments (optional):

Thanks for addressing the comments of the reviewer -- I'm happy with the results and your response to the referee. At this point, the paper is ready for acceptance. Congratulations!
---

## [Editor Report · Acceptance letter]

13 Jan 2022

PONE-D-21-03045R2 

Can big data increase our knowledge of local rental markets? a dataset on the rental sector in France 

Dear Dr. Chapelle:

I'm pleased to inform you that your manuscript has been deemed suitable for publication in PLOS ONE. Congratulations! Your manuscript is now with our production department. 

Kind regards, 

on behalf of

Dr. Nils Kok 

Academic Editor

PLOS ONE